# Associations between soil-transmitted helminth infections and physical activity, physical fitness, and cardiovascular disease risk in primary schoolchildren from Gqeberha, South Africa

Siphesihle Nqweniso[1]*, Cheryl Walter[1], Rosa du Randt[1], Larissa Adams[1], Johanna Beckmann[2], Jean T. Coulibaly[3,4,5,6], Danielle Dolley[1], Nandi Joubert[2,3,4], Kurt Z. Long[3,4], Ivan Müller[2], Madeleine Nienaber[1], Uwe Pühse[2], Harald Seelig[2], Peter Steinmann[3,4], Jürg Utzinger[3,4], Markus Gerber[2], Christin Lang[2]

1 Department of Human Movement Science, Nelson Mandela University, Gqeberha, South Africa,
2 Department of Sport, Exercise and Health, University of Basel, Basel, Switzerland, 3 Swiss Tropical and Public Health Institute, Allschwil, Switzerland, 4 University of Basel, Basel, Switzerland, 5 Centre Suisse de Recherches Scientifiques en Côte d'Ivoire, Abidjan, Côte d'Ivoire, 6 Unité de Formation et de Recherche Biosciences, Université Félix Houphouët-Boigny, Abidjan, Côte d'Ivoire

* fnqweniso@mandela.ac.za

**Data Availability Statement:** All data are in the manuscript and/or supporting information files.

## Abstract

### Background/Aim

School-aged children in low- and middle-income countries carry the highest burden of intestinal helminth infections, such as soil-transmitted helminths (STH). STH infections have been associated with negative consequences for child physical and cognitive development and wellbeing. With the epidemiological transition and rise in cardiovascular disease (CVD), studies have shown that helminth infections may influence glucose metabolism by preventing obesity. Thus, the aim of this study was to determine the association of STH infections in schoolchildren from Gqeberha, focusing on physical activity, physical fitness, and clustered CVD risk score.

### Methods

This cross-sectional study involved 680 schoolchildren (356 girls and 324 boys; mean age 8.19 years, SD±1.4) from disadvantaged communities in Gqeberha (formerly, Port Elizabeth), South Africa. Stool samples were collected and examined for STH infections using the Kato-Katz method. Physical activity (accelerometer) and physical fitness (grip strength, 20 m shuttle run) were measured using standard procedures. Furthermore, anthropometry, blood pressure, as well as glycated haemoglobin and lipid profile from capillary blood samples were assessed. We employed one-way ANOVAs to identify the associations of STH infections in terms of species and infection intensity with physical activity, physical fitness, and clustered CVD risk score.

**Funding:** MG was funded by the Fondation Botnar (Basel, Switzerland; project number 6071 'Physical activity and multi-micronutrient supplementation'), covering research expenses, staff salaries, study equipment and laboratory analyses. SN was funded by the Department of Research Development, Nelson Mandela University and the German Academic Exchange Service-National Research Foundation, South African (DAAD-NRF); Grant UID 117629. The funders played no role in the study design, data collection and data analysis, data interpretation, and preparation of the manuscript or decision to publish.

**Competing interests:** The authors have declared that no competing interests exist.

## Results

We found a low STH infection prevalence (7.2%) in our study, with participants infected with at least one intestinal helminth species. In comparison to their non-infected peers, children infected with STH had lower mean grip strength scores, but higher mean $VO_2$max estimation and higher levels of MVPA (p < .001). When considering type and intensity of infection, a positive association of *A. lumbricoides* infection and MVPA was found. In contrast, light *T. trichiura*-infected children had significantly lower grip strength scores compared to non and heavily-infected children. $VO_2$max and MVPA were positively associated with light *T. trichiura* infection. No significant association between the clustered CVD risk score and infection with any STH species was evident.

## Conclusions

STH-infected children had lower grip strength scores than their non-infected peers, yet, achieved higher $VO_2$max and MVPA scores. Our study highlights that the type and intensity of STH infection is relevant in understanding the disease burden of STH infections on children's health. The findings of our study must be interpreted cautiously due to the low infection rate, and more research is needed in samples with higher prevalence rates or case-control designs.

### Author summary

South Africa is experiencing a change in lifestyle and traditional dietary habits, causing a growing burden of non-communicable diseases (diabetes, obesity and cardiovascular diseases). In addition to this growing burden of non-communicable diseases, communicable diseases (parasitic worm infections) still exist, especially in historically marginalised communities characterised by poverty, high unemployment, and lack of access to adequate sanitation and clean water. These communities foster the transmission and increase the risk of soil-transmitted helminth infections, especially in schoolchildren. A total of 680 schoolchildren aged 6–12 years were examined for soil-transmitted helminth infections, physical activity, physical fitness and cardiovascular disease risk. The analyses showed that being infected was associated with lower grip strength scores. Surprisingly, infected children had higher cardiorespiratory fitness and MVPA. We observed no association between being infected and the clustered cardiovascular disease risk score. We observed a positive association between *A. lumbricoides* infection and MVPA. While *T. trichiura* was negatively associated with grip strength and positively associated with cardiorespiratory fitness and MVPA. This study corroborated previous studies regarding low grip strength in infected children. Furthermore, our findings contributed new insights on the importance of the type and intensity of helminth infection in understanding the disease burden of helminth infections on children's health.

## Introduction

More than 1.5 billion of the global population are infected with soil-transmitted helminths (STH), with most of these infections occurring in children living in low- and middle-income

countries (LMICs) [1–3]. The most prevalent helminth species are *Ascaris lumbricoides (A. lumbricoides)*, *Trichuris trichiura (T. trichiura)*, and hookworms, and infection with more than one worm species is not uncommon [1]. According to the World Health Organization (WHO) [4], school-aged children in LMICs carry the highest burden of STH infections. The consequences of STH infections during childhood are manifold, including impaired growth, malnutrition, and delayed intellectual development and cognition [1,5].

In recent years, a growing body of research has investigated the effects of children's STH infection on physical fitness [6–10] and physical activity levels [11]. Physical activity is vital to young children's healthy physical and cognitive development. Later in life, regular physical activity becomes increasingly important in preventing non-communicable disease (NCD) risk factors [12]. Conversely, high physical activity levels may pose a risk factor for STH infection because it increases exposure to contaminated soil when playing and running outside regularly, especially in communities with poor water, sanitation, and hygiene (WASH). STH infection itself may then lead to increased morbidity and lower levels of physical fitness. Studies investigating the impact of STH infection on children's physical fitness have reported lower grip strength scores [9,10] and lower maximum oxygen uptake ($VO_2max$) [8]. In contrast, Hürlimann et al. [13] observed higher $VO_2max$ and standing broad jump scores in children infected with hookworm and *Schistosoma mansoni (S. mansoni)*. The authors attributed these findings to infected children's general higher activity patterns. In line with this notion, a study conducted in marginalised communities in Gqeberha (formerly Port Elizabeth), South Africa, revealed that infected children self-reported higher physical activity levels than their non-infected peers [11]. Especially water activities place children at an increased risk for schistosomiasis [13]. However, other cross-sectional studies conducted in Côte d'Ivoire [7] and South Africa [10] did not find an association between STH infection and $VO_2max$. The inconsistent findings may be explained by the fact that previous studies focused primarily on the prevalence of STH infection alone, without considering the type and intensity of infection.

An important factor in determining the morbidity of STH infections is infection intensity, which refers to the number of worms found in an individual [4,14]. In most individuals, light infections are not overtly symptomatic, but the symptoms progress as the infection gets more severe. Infection intensity is measured in eggs per gram (EPG) and categorised as light, moderate, or heavy [4]. Moderately and heavily infected individuals experience more health consequences than lightly-infected individuals [15]. Furthermore, hookworm infection is mainly associated with iron deficiency anaemia resulting from chronic gastrointestinal bleeding. Anaemia can also be a symptom of moderate to heavy infections with *T. trichiura*. In contrast, infections with *A. lumbricoides* do not exhibit these symptoms, suggesting that the type of STH infection may also be relevant to the observed impact on VO2max and overall disease burden [15].

While in LMICs, children continue to suffer from infectious diseases, such as STH, their parents are more likely to suffer from NCDs. There has been a long-standing notion that NCDs, such as cardiovascular diseases (CVDs), type 2 diabetes, hypertension, and obesity, are the diseases of western high-income countries [16]. However, current epidemiological data indicate that NCDs, and CVD risk factors in particular, now disproportionately affect people in LMICs, where more than three-quarters of global NCD deaths occur [16]. To what extent childhood STH infection may function as a potential driver or prevention factor for CVD risk factors is still understudied. Previous research has focused on adult populations [17–19] or animals [20], with only a few studies conducted on children and adolescents [21]. The studies conducted on humans have shown an association between *S. mansoni* and reduced risk of diabetes and CVD risk [17,18] and reported that chronic STH infection is associated with lower insulin resistance, independent of body mass index (BMI) [19]. Based on systematic reviews among adult populations, helminth infections may offer a protective mechanism through the

helminth-induced modulation of inflammatory pathways responsible in the development of insulin resistance, thereby preventing diabetes and improving insulin sensitivity [22,23].

Taken together, CVD risk factors are characterised by multiple overlapping lifestyles and clinical risk factors (e.g., obesity, diabetes or impaired glucose tolerance, physical inactivity, hypertension), which the WHO Global Health Observatory data have identified as common and preventable risk factors that underlie most CVDs. Childhood physical activity is a preventive factor for various CVDs later in life. Yet, outdoor physical activities may provide a potential risk factor for STH infection among school-aged children. To date, little is known about the relationship between STH infections, habitual physical activity levels, physical fitness, and CVD risk factors in primary schoolchildren living in marginalised areas. Therefore, the objective of this study is threefold: first, to describe the prevalence, type, and intensity of STH infections in four schools in Gqeberha, South Africa. Second, to identify whether STH-infected and non-infected children differ regarding their physical fitness, physical activity levels and clustered CVD risk score. Third, determine whether STH infection type and intensity are associated with physical fitness, habitual physical activity and clustered CVD risk score. Based on previous research, three hypotheses were formulated. Higher STH infection prevalence in children from densely populated and poor infrastructure areas (Hypothesis 1) (9). STH-infected children will have lower physical fitness and physical activity levels and lower clustered CVD risk compared to their non-infected peers (Hypothesis 2) (9, 19). High *A. lumbricoides* and *T. trichiura* infection intensity is associated with lower physical fitness, physical activity and lower clustered CVD risk compared to light and moderate intensity (Hypothesis 3) [8,21].

## Methods

### Ethics statement

Approval of the study was obtained from the responsible ethics committees at the Nelson Mandela University in Gqeberha, South Africa (reference number: H19-HEA-HMS-001) and the 'Ethikkommission Nordwest- und Zentralschweiz' in Switzerland (EKNZ; reference number: Req-2018-00608). Permission was also sought from the Eastern Cape Department of Education and Department of Health, South Africa. The KaziAfya intervention study was registered on August 9, 2018, with ISRCTN (http://www.isrctn.com/ISRCTN29534081) and was conducted following the study protocol [24], the principles of the Declaration of Helsinki and the guidelines of Good Clinical Practice (GCP) issued by the International Conference of Harmonisation (ICH). Formal written consent was obtained from the parents/guardians of the participants. All children submitted an informed consent form signed by their parent/guardian, giving verbal assent before being assessed. Children's participation was voluntary, and it was explained that withdrawal from the study was possible at any point without further obligation.

### Study design and location

This study presents cross-sectional baseline data from a larger cluster-randomised controlled trial (*KaziAfya*). The *KaziAfya* study aims to assess the effectiveness of physical activity and multi-micronutrient supplementation interventions on primary schoolchildren living in South Africa, Tanzania, and Côte d'Ivoire [24]. In the South African cohort of the study, baseline data collection was conducted from February to April 2019 in Gqeberha, Eastern Cape province. The study was conducted at four primary schools in socioeconomically disadvantaged communities inhabited by predominantly coloured (mixed-race) and black African children. The selected schools are categorised as quintile three schools. In South Africa, quintile three schools are situated in underprivileged communities and are non-fee-paying schools [25].

## Participants and procedure

Based on a priori power analysis (calculations based on G*power 3.1) [24], the final target sample comprised 1320 children per study site. Detailed information about sampling and selection of schools, study participants, management of helminth infections, and referrals can be found in the published study protocol [24]. In brief, participants were recruited from four peri-urban primary schools in Gqeberha, South Africa. The inclusion criteria included schools that; (i) were quintile three public schools, (ii) had facilities available to implement physical education classes, and (iii) did not participate in another research project or clinical trial. Children were included in the study if; (i) they were in grades 1–4 (aged 6–12 years) at baseline, (ii) their parent/guardian had signed the informed consent form, and (iii) they were not participating in other research projects at the same time as our study, and they did not have any adverse medical condition preventing participation in physical activity. All children submitted an informed consent form signed by their parent/guardian, giving verbal assent before being assessed. Children's participation was voluntary, and it was explained that withdrawal from the study was possible at any point without further obligation. Data collection was conducted by biokineticists, nurses and trained research assistants during school hours over two days.

## Measures

All procedures used in this study were based on standardised, validated and quality-controlled standard operating procedures [24].

## Anthropometric measurements

Body height was assessed using a Seca Stadiometer (Surgical SA; Johannesburg, South Africa), measured to the nearest 0.1 cm. With shoes off, children stood against the stadiometer, shoulders relaxed, back erect, and heels touching the stadiometer. Body weight (to nearest 0.1 kg) and percentage body fat (BF%) were assessed using bioelectrical impedance analysis (BIA) via an electronic scale (Tanita MC-580; Tanita Corp., Tokyo, Japan). Children were required to fast for 3 hours before data collection. With minimal clothing, children stood barefoot on the metal plates of the scale, allowing for optimal contact with the plates. After inputting height measurement on the Tanita scale, body mass index (BMI) could be read from the scale's monitor.

## Socioeconomic status (SES)

We assessed household SES using a parental survey, where parents/guardians had to answer questions related to several items about asset ownership (e.g. number of bedrooms, people per household). The items included and exact scoring to estimate SES have been described elsewhere [26]. Items possessed by >95% or <5% of the sample were excluded, with the remaining items used to generate a wealth index via principal component analysis (PCA). The use of PCA to generate a wealth index has been used and validated previously [27]. A high score reflected high SES; quintiles were built based on the wealth index, with quintiles 1 and 5 representing lowest/poorest and highest/wealthiest, respectively.

## Soil-transmitted helminth (STH) infection

A pre-labelled (unique ID and name) stool sample container was handed out to participants on the morning of data collection. The participants were instructed to return with the container filled with their morning stool the next day (at least 15 g). Sample collection was done in the morning, and containers were transported to a laboratory for diagnostic work-up within

the same day. For the detection of STH infection (*A. lumbricoides*, *T. Trichiura* and hookworms), the Kato-Katz technique was used [28]. For each sample, a duplicate 41.7 mg thick smear was prepared and read under a microscope by experienced laboratory technicians from Nelson Mandela University. The number of helminth eggs was counted and recorded for each species separately. According to WHO guidelines, EPG was used to calculate STH infection intensity (light, moderate, heavy) [4].

## Physical fitness

Two tests from the Eurofit physical fitness test battery were used [29], namely (i) grip strength test and (ii) 20 m shuttle run test. The grip strength test was used to assess upper body strength, and the 20 m shuttle run test was used to assess cardiorespiratory fitness [30]. Using the Saehan hydraulic hand dynamometer (MSD Europe BVBA; Tisselt, Belgium), children were tested on both right and left hands after adjusting the grip span. The test was conducted with the child seated upright, shoulders relaxed, and elbow bent at a 90-degree angle alongside the body. With alternating hands, six trials were conducted (with 30-sec rest in between). The final score was determined by calculating an average of all the trials (to the nearest 1 kg). The 20 m shuttle run test was conducted in the schoolyard on flat ground, adhering to a standard test protocol [30,31]. Children were instructed to run to a pre-recorded signal with an initial running speed of 8.5 km/h that increased by 0.5 km/h after every minute. The test was terminated when a child failed to follow the pace in two consecutive 20 m intervals. The last completed lap and corresponding speed were recorded and converted to maximal oxygen uptake ($VO_2$max) using the speed and age of each participant [30].

## Moderate-to-vigorous physical activity (MVPA)

Physical activity was assessed with an accelerometer device (ActiGraph wGT3X-BT; Pensacola, USA). Children were instructed to wear the device around the hip for seven consecutive days and to take it off only when in contact with water. Devices collected data from 6 a.m. to midnight, and data were deemed valid if children wore the device for at least 8 hours on four weekdays and one weekend day [32], and the accelerometer device (ActiGraph) had been validated for children [33]. Moderate and vigorous physical activity were determined using raw data and the ActiLife software version 6.13.2, with raw data set to 10-sec epochs [34]. Children-specific cut-off points were used to determine MVPA [35].

## Cardiovascular risk factors

An Omron M3 digital blood pressure monitor (Omron Healthcare Europe; Hoofddorp, The Netherlands) was used to measure blood pressure three times after the child had been resting for 5 minutes, with a 1-minute rest between measures. An average of the last two blood pressure measurements was used as the final reading. For the assessment of blood lipid profiles (high-density lipoprotein (HDL), triglycerides and total cholesterol to HDL ratio) and glycated haemoglobin (HbA1c) level, capillary blood was analysed using the Afinion test (Alere Technology, Abbot; Wädenswil, Switzerland). The finger prick technique was used to collect drops of blood, with children having fasted for 3 hours before the test.

## Statistical analysis

Data were double-entered, validated using EpiData version 3.1 (EpiData Association; Odense, Denmark) and merged into a single file. Descriptive statistics are reported as mean (M) and standard deviation (SD) or number (n) and percentage (%). One-way ANOVAs were used to

determine whether grip strength, VO$_2$max, MVPA and clustered CVD risk score differed between infected and non-infected children. Additional subgroup analyses were carried out at one school with the highest STH prevalence. Similarly, one-way ANOVAs were used to determine whether grip strength, VO$_2$max, MVPA, and clustered CVD risk scores differed by STH infection type and intensity. When the p-value revealed significant differences between some of the tested variables, a Tukey's Honest Significant Difference (HSD) test was used to assess the significant differences between pairs of group means. Possible group differences in children's sex, age and BMI are controlled for in the models. The VO$_2$max calculation already considers age [30]. To obtain a clustered cardiovascular risk score, the individual risk factors were z-standardized and added up using the following formula to calculate the clustered risk score: (systolic + diastolic blood pressure/2) + body fat + ratio of total cholesterol to HDL + triglycerides + HbA1c. Previous publications based on studies conducted in Europe, the United States of America [36,37], and South Africa, Gqeberha [38,39] have used this formula. SES was determined by calculating a wealth index generated through a PCA, with higher scores reflecting higher SES. Details about the wealth index calculation have been described elsewhere [26]. All statistical analyses were carried out using SPSS Version 27 (IBM Corporation, Armonk, NY, USA) for Windows. For all statistical analyses, we used p < 0.05 to denote significance.

## Results

### Sample characteristics

A total of 1369 children provided informed consent, with 65 dropping out before baseline assessment due to relocation or moving to a different school. Of the 1304 remaining children, 624 were not included in the data analysis as they had missing information for at least one of the following: age, sex, stool sample, grip strength, 20m shuttle run, accelerometer or clustered CVD score (Fig 1). Subsequent analyses were based on 680 children with complete data (356 girls, 324 boys), a mean age of 8.19 years (SD±1.4). There were significant age, sex and BMI differences between the four schools (p < 0.001); thus, these variables were included as covariates in all subsequent analyses. No SES differences were observed. Table 1 shows the descriptive statistics of all measured variables for the sample (n = 680). The mean grip strength, VO$_2$max estimate and MVPA scores for the study cohort were 11.40 kg, 47.56 ml/kg/min and 80.09 min/day, respectively.

Due to almost half of the study participants having missing data, we tested whether differences existed between children with complete vs. incomplete data, and we present this data as supplementary online material (S1 Table). We found significant group differences regarding age (8.40 vs. 8.20 years), height (125.32 vs. 123.98 cm), weight 26.23 vs. 24.72 kg), BMI (16.42 vs. 15.89 kg/m$^2$) and MVPA (82.64 vs. 80.07 min/day), with children with incomplete data having higher scores compared to those with complete data. The effect sizes for age, height, weight, BMI and MVPA, calculated as partial eta squared ($\eta^2$), were all between 0.005–0.011, indicating a small effect.

### Prevalence, type, and intensity of STH infection

Forty-nine out of 680 children (7.2%) were infected with at least one intestinal helminth species. Infection prevalence of *A. lumbricoides* and *T. trichiura* were 5.8% and 7.0%, respectively. Only one child was infected with hookworm, and no separate analyses were conducted for this child. Out of the 49 infected children, co-infection was found in 40 (81.6%) children who harboured both *A. lumbricoides* and *T. trichiura*. Stratification by school revealed that school 2 had the highest prevalence of both *A. lumbricoides* and *T. trichiura* (Table 1). Children infected with *A. lumbricoides* presented mostly with light infection intensity. In contrast, those infected

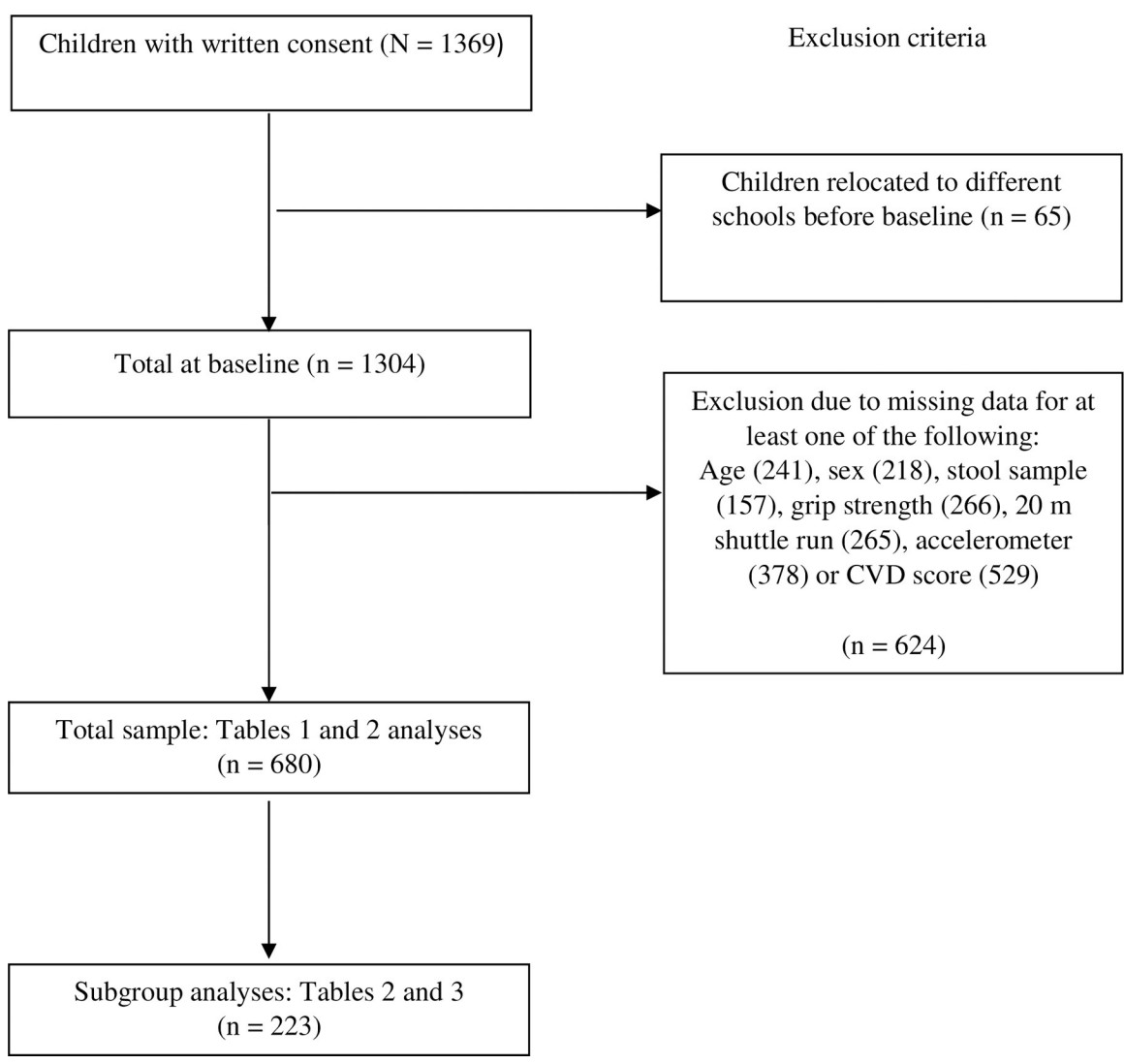

**Fig 1. Participant flow chart.**

with *T. trichiura* showed mostly heavy infection intensity, with no moderate infection intensity observed in *T. trichiura* infection.

### Physical fitness, MVPA and CVD risk among infected and non-infected children

Table 2 shows the total group and subgroup analyses of infected vs. non-infected children with regards to physical fitness, MVPA and CVD risk. As shown in Table 2, children infected with STH had significantly lower mean grip strength than their non-infected peers ($F(1,679) = 7.53$, $p = 0.006$, $\eta^2 = 0.011$). Mean $VO_2$max estimate was found to be significantly higher in infected compared to non-infected children ($F(1,679) = 22.06$, $p < 0.001$, $\eta^2 = 0.032$). Similar results were found for MVPA, with infected children achieving higher scores compared to non-infected children ($F(1,679) = 21.88$, $p < 0.001$, $\eta^2 = 0.031$). No significant differences between infected and non-infected children in respect of the clustered CVD risk score could be

**Table 1. Characteristics of study participants from marginalised communities in Gqeberha, South Africa (n = 680).**

| Parameters | N | Total | School 1 (n = 230) | School 2 (n = 223) | School 3 (n = 144) | School 4 (n = 83) | Statistics | | |
|---|---|---|---|---|---|---|---|---|---|
| | | M (SD) | M (SD) | M (SD) | M (SD) | M (SD) | F | p | $\eta^2$ |
| Age (years) | 680 | 8.19 (1.42) | 8.38 (1.48) | 8.24 (1.45) | 8.19 (1.42) | 7.59 (0.99) | 6.44 | < 0.001 | 0.028 |
| Height (cm) | 680 | 123.98 (9.08) | 126.48 (8.94) | 122.53 (9.25) | 123.55 (9.40) | 121.67 (6.72) | 9.98 | < 0.001 | 0.042 |
| Weight (kg) | 680 | 24.72 (6.13) | 27.25 (6.66) | 23.45 (5.67) | 23.38 (5.24) | 23.42 (4.76) | 21.57 | < 0.001 | 0.087 |
| BMI (kg/m$^2$) | 680 | 15.89 (2.35) | 16.87 (2.71) | 15.43 (2.07) | 15.16 (1.78) | 15.69 (1.95) | 23.10 | < 0.001 | 0.093 |
| **SES** | 680 | 3.07 (1.44) | 3.13 (1.49) | 3.03 (1.44) | 3.17 (1.39) | 2.88 (1.36) | 0.73 | 0.535 | 0.005 |
| **STH infection** | N | N (%) | N (%) | N (%) | N (%) | N (%) | $\chi^2$ | p | $\eta^2$ |
| *A. lumbricoides* | 680 | 40 (5.8) | 0 (0.0) | 39 (5.7) | 1 (0.1) | 0 (0.0) | 80.45 | < 0.001 | 0.344 |
| *T. trichiura* | 680 | 48 (7.0) | 0 (0.0) | 47 (6.9) | 1 (0.1) | 0 (0.0) | 98.99 | < 0.001 | 0.381 |
| Hookworm | 680 | 1 (0.1) | 0 (0.0) | 0 (0.0) | 0 (0.0) | 1 (0.1) | 7.22 | 0.065 | 0.103 |
| Single infection | 680 | 9 (1.3) | 0 (0.0) | 8 (1.2) | 0 (0.0) | 1 (0.1) | – | – | – |
| Co-infection | 680 | 40 (5.8) | 0 (0.0) | 39 (5.7) | 1 (0.1) | 0 (0.0) | – | – | – |
| **Physical fitness** | | M (SD) | M (SD) | M (SD) | M (SD) | M (SD) | F | p | $\eta^2$ |
| Grip strength (kg) | 680 | 11.40 (4.87) | 10.94 (4.23) | 10.50 (3.48) | 13.86 (7.12) | 10.83 (3.45) | 16.96 | < 0.001 | 0.070 |
| VO$_2$max (ml/kg/min) | 680 | 47.56 (3.48) | 46.79 (3.14) | 48.84 (3.78) | 46.07 (2.80) | 48.88 (2.93) | 30.09 | < 0.001 | 0.118 |
| **MVPA (min/day)** | 680 | 80.09 (26.39) | 73.75 (24.82) | 84.14 (27.05) | 82.34 (26.21) | 82.86 (26.32) | 7.01 | < 0.001 | 0.030 |
| **Clustered CVD risk** | 680 | -0.04 (2.68) | 0.16 (2.75) | -0.14 (2.66) | -0.51 (2.56) | 0.45 (2.68) | 2.98 | 0.031 | 0.013 |

Note. BMI = body mass index, SES = socioeconomic status, STH = Soil-transmitted helminth infection, VO$_2$max = Cardiorespiratory fitness, MVPA = Moderate-to-vigorous physical activity, CVD = Cardiovascular disease, $\eta^2$ = partial eta squared: 0.01 = small, 0.06 = medium, 0.14 = large effect size.

established (F(1,679) = 0.178, p > 0.05, $\eta^2$ = 0.000). After controlling for sex, age and BMI, the differences in grip strength, VO$_2$max and MVPA remained statistically significant, with the clustered CVD risk score still showing no significant difference between infected and non-infected children.

Due to infection occurring in one of the four schools under investigation, further subgroup analyses were carried out, focusing on the high infection area school (school 2, n = 223). The

**Table 2. Physical fitness, physical activity (MVPA), and clustered CVD risk in non-infected and STH-infected children (n = 680).**

| | Total sample analyses | | | | | | | | |
|---|---|---|---|---|---|---|---|---|---|
| Measures | Non-infected (n = 631) | STH-infected (n = 49) | Unadjusted model | | | Model adjusted for sex, age and BMI | | | |
| | M (SD) | M (SD) | F | p | $\eta^2$ | F | p | $\eta^2$ | |
| Grip strength (kg) | 11.55 (4.97) | 9.57 (2.97) | 7.53 | **0.006** | 0.011 | 6.55 | **0.011** | 0.010 | |
| VO$_2$max (ml/kg/min) | 47.39 (3.36) | 49.78 (4.16) | 22.06 | **<0.001** | 0.032 | 18.71 | **<0.001** | 0.027 | |
| **MVPA (min/day)** | 78.79 (26.18) | 96.82 (23.34) | 21.88 | **<0.001** | 0.031 | 17.34 | **<0.001** | 0.025 | |
| **Clustered CVD risk** | -0.03 (2.71) | -0.20 (2.32) | 0.178 | 0.674 | 0.000 | 1.07 | 0.301 | 0.002 | |

| | Subgroup analyses | | | | | | | | |
|---|---|---|---|---|---|---|---|---|---|
| Measures | Non-infected (n = 176) | STH-infected (n = 47) | Unadjusted model | | | Model adjusted for sex, age and BMI | | | |
| | M (SD) | M (SD) | F | p | $\eta^2$ | F | p | $\eta^2$ | |
| Grip strength (kg) | 10.77 (3.55) | 9.50 (3.01) | 4.99 | **0.026** | 0.022 | 5.32 | **0.022** | 0.024 | |
| VO$_2$max (ml/kg/min) | 48.55 (3.64) | 49.93 (4.13) | 5.079 | **0.025** | 0.022 | 2.75 | 0.099 | 0.012 | |
| **MVPA (min/day)** | 80.81 (26.96) | 96.59 (23.78) | 13.32 | **<0.001** | 0.057 | 9.33 | **0.003** | 0.041 | |
| **Clustered CVD risk** | -0.16 (2.76) | -0.08 (2.25) | 0.034 | 0.854 | 0.000 | 1.51 | 0.221 | 0.007 | |

Note. STH = Soil-transmitted helminth infection, BMI = Body mass index, VO$_2$max = Cardiorespiratory fitness, MVPA = Moderate-to-vigorous physical activity, CVD = Cardiovascular disease, $\eta^2$ = Partial eta squared: 0.01 = small, 0.06 = medium, 0.14 = large effect size.

subgroup analyses indicates that STH-infected children showed significantly lower mean grip strength and higher $VO_2max$ compared to non-infected children ($F(1,222) = 4.99$, $p = 0.026$, η2 = 0.022) and ($F(1,222) = 5.08$, $p = 0.025$, η2 = 0.022), respectively. Mean MVPA scores were significantly higher for infected children compared to their non-infected peers ($F(1,222) = 13.32$, $p < 0.001$, η2 = 0.057). These results remained significant even after adjusting for covariates (sex, age and BMI), except for $VO_2max$. No significant differences were found between infected and non-infected children regarding the clustered CVD risk score ($F(1,222) = 0.03$, $p > 0.05$, η2 = 0.000).

## Association of infection type and intensity with physical fitness, MVPA and CVD risk

When considering STH infection type and intensity in the subgroup analyses (Table 3), infection with *A. lumbricoides* showed no significant overall effect with regard to infection intensity and grip strength and $VO_2max$ ($F(3,220) = 0.59$, $p = 0.621$, $η^2 = 0.008$) and ($F(3,220) = 0.81$, $p = 0.492$, $η^2 = 0.011$), respectively. There was a significant overall effect with regards to infection intensity and MVPA ($F(3,220) = 3.38$, $p = 0.019$, $η^2 = 0.044$). However, the association between *A. lumbricoides* infection intensity and MVPA was not observed when the model was adjusted for sex, age and BMI ($F(3,220) = 2.04$, $p = 0.109$, $η^2 = 0.028$). A post hoc Tukey test showed no significant differences between infection intensities ($p > 0.05$). Infection intensity did not show any significant overall effects on clustered CVD risk scores.

Furthermore, subgroup analyses revealed that *T. trichiura* infection intensity had a significant overall effect on grip strength ($F(3,220) = 5.14$, $p = 0.007$, $η^2 = 0.045$). A post hoc Tukey test indicated a significant difference between non and light infection ($p = 0.007$). There was a significant overall effect for infection intensity and $VO_2max$, in that $VO_2max$ increases from non to light infection yet, decreases again for heavy infections $F(3,220) = 4.74$, $p = 0.010$, $η^2 = 0.041$), with no such significance observed in the adjusted model. Again, a post hoc test showed a significant difference between non and light infection ($p = 0.011$). Similar to $VO_2max$,

**Table 3. Sub-analyses: Physical fitness, physical activity (MVPA), and clustered CVD risk in non-infected versus STH-infected children, separately for type (*A. lumbricoides* and *T. trichiura*) and intensity (light vs. moderate vs. heavy as determined by EPG) (school 2: n = 223).**

| | *Ascaris lumbricoides* | | | | | | | | | |
|---|---|---|---|---|---|---|---|---|---|---|
| | Non-infected (n = 184) | Light (n = 16) | Moderate (n = 11) | Heavy (n = 12) | Unadjusted model | | | Model adjusted for sex, age and BMI | | |
| Parameters | M (SD) | M (SD) | M (SD) | M (SD) | F | p | $η^2$ | F | p | $η^2$ |
| Grip strength (kg) | 10.60 (3.56) | 10.13 (3.16) | 10.64 (2.26) | 9.31 (3.29) | 0.59 | 0.621 | 0.008 | 0.99 | 0.401 | 0.013 |
| VO₂max (ml/kg/min) | 48.72 (3.74) | 50.03 (3.94) | 48.36 (4.55) | 49.59 (3.59) | 0.81 | 0.492 | 0.011 | 0.55 | 0.647 | 0.008 |
| MVPA (min/day) | 81.64 (27.00) | 96.39 (21.66) | 100.97 (30.95) | 90.69 (21.85) | 3.38 | **0.019** | 0.044 | 2.04 | 0.109 | 0.028 |
| Clustered CVD risk | -0.21 (2.73) | 0.17 (2.67) | -0.76 (2.18) | 0.80 (1.82) | 0.80 | 0.497 | 0.011 | 1.10 | 0.349 | 0.015 |

| | *Trichuris trichiura* | | | | | | | | | |
|---|---|---|---|---|---|---|---|---|---|---|
| | Not-infected (n = 176) | Light (n = 8) | Moderate (n = 0) | Heavy (n = 39) | Unadjusted model | | | Model adjusted for sex, age and BMI | | |
| Parameters | M (SD) | M (SD) | M (SD) | M (SD) | F | p | $η^2$ | F | p | $η^2$ |
| Grip strength (kg) | 10.77 (3.55) | 7.00 (1.91) | – | 10.02 (2.95) | 5.14 | **0.007** | 0.045 | 3.22 | **0.042** | 0.029 |
| VO₂max (ml/kg/min) | 48.55 (3.64) | 52.42 (4.19) | – | 49.42 (3.98) | 4.74 | **0.010** | 0.041 | 2.20 | 0.113 | 0.020 |
| MVPA (min/day) | 80.81 (26.90) | 99.83 (22.14) | – | 95.93 (24.33) | 6.71 | **0.001** | 0.057 | 4.83 | **0.009** | 0.043 |
| Clustered CVD risk | -0.17 (2.76) | -0.97 (1.69) | – | 0.10 (2.32) | 0.56 | 0.574 | 0.005 | 1.01 | 0.367 | 0.009 |

Note. STH = Soil-transmitted helminth infection, BMI = Body mass index, VO₂max = Cardiorespiratory fitness, MVPA = Moderate-to-vigorous physical activity, CVD = Cardiovascular disease, EPG = Eggs per gram, $η^2$ = Partial eta squared: 0.01 = small, 0.06 = medium, 0.14 = large effect size.

MVPA and infection intensity with *T. trichiura* showed a significant overall effect F(3,220) = 6.71, p = 0.001, $\eta^2$ = 0.057), with post hoc test showing a significant difference between non and heavy infection (p = 0.004). We did not establish significant differences between the clustered CVD risk score and *T. trichiura* infection intensity (p > 0.05).

## Discussion

The present study investigated the prevalence and intensity of STH infection and its association with selected health parameters in primary schoolchildren from disadvantaged communities in Gqeberha, South Africa. A key finding was that the prevalence of STH infection was highly dependent on the geographic location of schools. Overall, STH-infected children had lower grip strength scores than their non-infected peers yet, achieved higher VO₂max and MVPA scores. Subgroup analyses also showed significantly lower grip strength and higher VO₂max and MVPA scores in infected children compared to their non-infected peers. Furthermore, in the subgroup, *A. lumbricoides* infection intensity revealed no differences in grip strength, VO₂max and CVD risk score but showed higher MVPA among infected children, independent of infection intensity. In contrast, for *T. trichiura* infection, lightly-infected children had significantly lower grip strength and achieved higher VO₂max estimates. While heavily-infected children had significantly higher MVPA levels than their light and non-infected peers. There was an unexpected reversal in this pattern for grip strength, with non-infected and *T. trichiura* heavily-infected children having higher grip strength scores compared to their lightly-infected peers; however, this may be due to the low number of lightly-infected children (n = 8). In contrast, clustered CVD risk score was not associated with STH infection intensity.

The present study had three hypotheses, each of which will now be discussed. Firstly, we hypothesised that a higher STH prevalence would be observed in areas that are densely populated and have poor infrastructure. Overall, our study found a low prevalence of *T. trichiura* and/or *A. lumbricoides* infection (7.2%), with only one child infected with hookworms. The STH infection prevalence is much lower than what has been reported in previous studies in Gqeberha (31.0%) [10] and KwaZulu-Natal (1.2–26%) [40,41]. The low prevalence can be attributed to a national school deworming programme launched in 2016 by the South African Department of Education for schools in disadvantaged communities [42]. Müller and colleagues [43] also reported shrinking risk profiles following three rounds of albendazole deworming of STH-infected children from communities located in close proximity to the current study setting (March 2015, October 2015 & May 2016). Even though the general prevalence in our sample population was found to be low, one school showed substantially higher STH infection compared to the other three schools. This observation is in line with findings from the DASH study that also reported high variations between schools located within similar geographical areas [9]. Their study found that two schools situated in the same geographical area as the present study had high infection rates. This geographical area is one of the old apartheid group areas set aside for coloured people during the period of forced removals in the 1960s [44,45], and it is characterised by high population density and inadequate water and sanitation facilities to cope with the high density of people living in the area [9]. The present study revealed that 81.6% of the infected children were co-infected with *A. lumbricoides* and *T. trichiura*. This is higher than the findings of previous studies in China and South Africa [8,9,41], where 30–51% of their participants harboured both *A. lumbricoides* and *T. trichiura*, which is common due to the mechanism of infection [14].

Secondly, we hypothesised that STH-infected children would have lower physical fitness and physical activity levels and lower clustered CVD risk compared to their non-infected

peers. These results are reported for both the whole group (n = 680) and subgroup (n = 223). Consistent with existing literature, STH-infected children in the whole sample and subgroup had a lower grip strength score than their non-infected peers [8,9]. Interestingly, STH-infected children had significantly higher $VO_2$max than non-infected children. These findings contradict previous research conducted in China and South Africa, which found lower mean $VO_2$max estimates in STH-infected children [8,9], but corroborate findings from Gerber and colleagues [11] where self-reported physical activity was associated with a higher infection rate in children from a similar area. A possible reason for the high $VO_2$max scores in the current study is that infected children have significantly lower BMI than non-infected children, allowing them to complete more laps on the 20 m shuttle run test. High BMI in children has been associated with poor performance in weight-bearing activities such as the 20 m shuttle run test [46]. This was also found to be the case in a study by Smith and colleagues [47], who reported that overweight/obese children completed fewer laps in the 20 m shuttle run test and had poor $VO_2$max scores. Similar to the $VO_2$max results, STH-infected children in the whole group and subgroup had significantly higher MVPA scores. These observations may be explained by the fact that physically active children are more likely to play outside (sometimes barefoot), which may place them at a higher risk of contracting infection, especially in areas with poor water and sanitation. A previous study in KwaZulu-Natal found a positive association between schistosomiasis and physical fitness. The authors attributed this to a selective increase in exposure to infected water in more active children [48]. Although a physically active lifestyle from a young age onwards is a key component in preventing NCDs later in life, physical activity guidelines need to address risk factors in disadvantaged areas. The importance of regular deworming in schoolchildren has been underscored by the WHO, as it ensures the reduction in morbidity caused by the infection and the occurrence of severe complications [4]. On the other hand, previous research suggests a protective mechanism of STH infection against metabolic outcomes. For instance, *S. mansoni* infection was positively associated with blood lipids (lower mean low-density lipoprotein, total cholesterol, and triglycerides) and diastolic blood pressure [17,18]. Another study conducted in Flores Island, a highly STH-endemic area in Indonesia, found that STH infections lowered insulin resistance and that infection with an increase in STH species further increased insulin sensitivity in adults [19]. However, in our study, we could not establish such a relationship. However, we recommend that future studies investigate the possible link between STH infection and CVD risk factors in children.

Thirdly, we hypothesised that high *A. lumbricoides* and *T. trichiura* infection intensity is associated with lower physical fitness, physical activity and lower clustered CVD risk compared to light and moderate intensity. To our knowledge, previous studies have not considered the type and intensity of infection when investigating the effects of STH infections on children's health, especially physical activity, physical fitness and clustered CVD risk factors. We did not find any significant effect with regard to *A. lumbricoides* infection intensity and grip strength, and $VO_2$max. A study conducted in China where *A. lumbricoides* infected children had lower grip strength scores revealed increases in grip strength following one month of deworming treatment with albendazole [49]. We also found that *A. lumbricoides* infected children had significantly higher MVPA compared to their non-infected counterparts; however, Tukey's HSD post hoc test did not show any significant differences between infection intensity. In our findings, light *T. trichiura* infection intensity resulted in significantly lower grip strength and higher $VO_2$max, which does not corroborate previous studies that reported a decrease in $VO_2$max of infected children [9]. However, a study in China showed no clear relationship between infection intensity and physical fitness [8]. We also found significantly higher MVPA in *T. trichiura*-infected children, with the post hoc test indicating significant differences between non and highly-infected children. Our findings show a trend towards an

inverse U-shaped relationship for *T. trichiura* infection, in that lightly-infected children achieved higher VO$_2$max levels compared to their heavily-infected and non-infected peers. However, we reported low light *T. trichiura* infection (n = 8) compared to heavy intensity (n = 39), indicating an imbalanced portion of children with different infection intensities. This means we do not have sufficient portions of different infection intensities to confirm these results, but we can only assume this inverse relationship, and our data shows some support for this relationship. Due to our low infection prevalence and low numbers of light infection intensity, more research in samples with higher infectious rates or with case-control design is needed. Light STH infections usually have little to no long-term health consequences, whereas moderate and high infection intensity results in a high disease burden [1]. Consequently, a light infection may pose an advantage in weight-bearing activities due to lower BMI, but this is outweighed by increasing infection intensity and hence, an increase in disease burden. Low BMI in *T. trichiura*-infected children has been reported in previous studies [8,9] due to hindered nutrient uptake, and light infection may also contribute to this, especially in cases of recurring infection [1,14]. Previous studies investigating the effects of STH infections on nutrition status and physical fitness reported that being anaemic and stunted had a negative impact on children's physical fitness [50]. Anaemia is a common symptom of hookworm infection, and it can also be a symptom of moderate to heavy infection with *T. trichiur*a, while *A. lumbricoides* do not exhibit these symptoms [15]. Since hookworm has the most detrimental effects on children's physical fitness due to iron deficiency, anaemia and malnutrition [14,50], and as we only observed one case of hookworm infection, we cannot make conclusions on the fitness benefits seen in STH-infected children.

## Limitations and strengths

Due to the study's cross-sectional nature, we could not report inferences about causality and its direction. The overall sample size was small, and STH infection prevalence was low in our population. Our findings cannot be generalised to the entire South African population as children from only one province, and social strata were included in the investigation. We only collected one stool sample, whereas WHO recommends the collection of three stool samples per participant for accurate detection of infection. Furthermore, our diagnostic method has limitations regarding its ability to pick up low-intensity infection [15]. Therefore, a concentration test with higher sensitivity is recommended for future studies. Apart from the limitations identified in the study, some strengths can be noted, such as the study's novelty in investigating the type and intensity of infection among children in Gqeberha, South Africa. Our results highlight the importance of focusing on the type and intensity of STH infections to determine disease morbidity, as prevalence alone cannot display a complete picture of the disease burden. Most of the prevalence of infection observed in one area and the differences amongst infected and non-infected children regarding VO$_2$max and MVPA, which have not been reported elsewhere, warrant more extensive follow-up studies.

## Conclusions

Our data suggest that STH infections hamper children's upper body strength. We also highlighted that there are STH infection hotspot areas in Gqeberha, which warrants regular deworming programmes and WASH interventions. Our findings revealed no clear relationship between STH infection and CVD risk factors. Hence, future research should seek a deeper understanding of the mechanism of how helminths mediate metabolic pathways, especially in children. Our study highlights that the type and intensity of STH infection are relevant in understanding the disease burden of STH infections on children's health. We hope that more

empirical evidence will emerge from future studies with larger samples, from areas where the authorities have not yet been able to perform systematic deworming and where infection rates are likely to be higher, and/or with case-control designs with a balanced portion of children with different infection intensity.

## Supporting information

**S1 Table. Differences between children with complete vs. incomplete data.**
(DOCX)

## Acknowledgments

The authors are grateful to the school authorities, school staff, participating children, and their parents/guardians for providing consent and assent to participate in the study. A special thanks to Zaahira Ismail for her administrative work and contribution to data collection. We are also thankful and appreciative to all the postgraduate students who participated and dedicated their time to data collection and data entry. We are grateful to the team of the Department of Medical Laboratory Sciences at the Nelson Mandela University for providing diagnostic support in the laboratory. In-kind contributions are provided by all involved parties. This research took place under the auspices of the UNESCO Chair on "Physical Activity and Health in Educational Settings" (https://unesco-chair.dsbg.unibas.ch/en/).

## Author Contributions

**Conceptualization:** Siphesihle Nqweniso, Johanna Beckmann, Jean T. Coulibaly, Markus Gerber, Christin Lang.

**Formal analysis:** Siphesihle Nqweniso.

**Funding acquisition:** Cheryl Walter, Kurt Z. Long, Uwe Pühse, Harald Seelig, Peter Steinmann, Jürg Utzinger, Markus Gerber, Christin Lang.

**Investigation:** Siphesihle Nqweniso, Larissa Adams, Johanna Beckmann, Danielle Dolley, Nandi Joubert, Ivan Müller, Madeleine Nienaber.

**Methodology:** Siphesihle Nqweniso, Christin Lang.

**Project administration:** Siphesihle Nqweniso, Kurt Z. Long, Markus Gerber, Christin Lang.

**Supervision:** Cheryl Walter, Rosa du Randt, Markus Gerber, Christin Lang.

**Validation:** Markus Gerber, Christin Lang.

**Visualization:** Siphesihle Nqweniso.

**Writing – original draft:** Siphesihle Nqweniso.

**Writing – review & editing:** Siphesihle Nqweniso, Cheryl Walter, Rosa du Randt, Larissa Adams, Johanna Beckmann, Jean T. Coulibaly, Danielle Dolley, Nandi Joubert, Kurt Z. Long, Ivan Müller, Madeleine Nienaber, Uwe Pühse, Harald Seelig, Peter Steinmann, Jürg Utzinger, Markus Gerber, Christin Lang.

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
