## [Decision Letter · Decision Letter 0]

13 Mar 2023

Dear Ms Nqweniso,

Thank you very much for submitting your manuscript "Influence of Soil-transmitted helminth infections on physical activity, physical fitness, and cardiovascular disease risk in primary schoolchildren from Gqeberha, South Africa" for consideration at PLOS Neglected Tropical Diseases. As with all papers reviewed by the journal, your manuscript was reviewed by members of the editorial board and by several independent reviewers. In light of the reviews (below this email), we would like to invite the resubmission of a significantly-revised version that takes into account the reviewers' comments. 

We cannot make any decision about publication until we have seen the revised manuscript and your response to the reviewers' comments. Your revised manuscript is also likely to be sent to reviewers for further evaluation.

Sincerely,

Sabine Specht

Academic Editor

Eva Clark

Section Editor

Reviewer's Responses to Questions

**Key Review Criteria Required for Acceptance?**

**Methods**

-Are the objectives of the study clearly articulated with a clear testable hypothesis stated?

-Is the study design appropriate to address the stated objectives?

-Is the population clearly described and appropriate for the hypothesis being tested?

-Is the sample size sufficient to ensure adequate power to address the hypothesis being tested?

-Were correct statistical analysis used to support conclusions?

-Are there concerns about ethical or regulatory requirements being met?

Reviewer #1: The objectives of the study are clear. Power calculations are for the original study and this analysis is just from the baseline data. No concerns with ethics or regulatory. Measurement of variables of interest seem appropriate. In the intro, the causal pathway of physical activity, CVD risk and STH infection is not clear, which would change which variables are exposures versus outcomes. For the purpose of this study, a hypothesis could be stated to help the reader understand why the analysis is being chosen as it is. Based on the analysis, the STH infection is being treated as an exposure. If infection is only being evaluated as a dichotomous variable, why not use a chi-square test rather than an ANOVA?

Reviewer #2: There are some questions regarding the method:

1. In comparison to their non-infected peers, children infected with STH had lower mean grip strength scores, but higher mean VO2max estimation and higher levels of MVPA. 

2. Why did the authors choose this measurement? Would it be the lifestyle that cause this difference? Can the infected children have more outdoor activities than the non-infected as mentioned?

3. Is the lower grip strength score associated with energy intake or excessive energy uses?

4. Any information collected on their habits or lifestyle beside the MVPA?

5. The authors have discussed this later in the discussion part, however if they had the data on characteristics of the physical activities of the school and the non-infected children, that might be useful information. If those children have similar physical activities levels, then it might be interesting to understand why they showed different results than other study.

6. More than 80% of the infected children has coinfection. Any plan on testing the stools with a more sensitive measurement such as PCR? 

7, The VOmax and the MVPA results seem to show different between schools. Is there any characteristics information collected on the schools? The authors indeed did an analyses in one school with relatively high helminth prevalence. But I wonder whether the school itself and the children went to this school also had different characteristic than other schools.

**Results**

-Does the analysis presented match the analysis plan?

-Are the results clearly and completely presented?

-Are the figures (Tables, Images) of sufficient quality for clarity?

Reviewer #1: With almost half the cohort being dropped for missing data, some description of that group would be helpful to understand how missing data may be biasing the results. Table 1 by school is confusing as there is almost no STH in the schools except one. Is there a more informative way to structure this table? Many of the outcomes vary by age. How do the outcomes vary by gender. Is it shown how age and BMI are associated with STH infection? Table 1 by STH infection (yes/no) excluding your outcome variables would help indicate why and which variables you are treating as confounders. 

For table 2, adjusted and unadjusted outcomes in one table would be helpful for comparison. And what is adjusted for, would relate to what is seen in table 1 as being associated with STH infection. Because STH infection numbers are so small, differences by intensity are even smaller. Did you consider combining light and moderate?

Reviewer #2: Yes.

**Conclusions**

-Are the conclusions supported by the data presented?

-Are the limitations of analysis clearly described?

-Do the authors discuss how these data can be helpful to advance our understanding of the topic under study?

-Is public health relevance addressed?

Reviewer #1: Drawing conclusions from intensity of infection and outcomes is challenged by small numbers. Focusing on conclusions based on infected or not may be more informative. In the paragraph starting on line 398, the causal pathway of physical activity, CVD and STH infection is hard to follow. The first sentence of the limitations states that the direction of the causality can not be inferred due to the cross sectional study design. I agree but the analysis is based on an assumption so interpreting the results aligned with the hypothesis of the study may help with clarity.

Reviewer #2: The authors have addressed the conclusion, the limitation and the strengths of the study, however they might want to explore studies related to the topic they raised such as studies done in Indonesia.

**Editorial and Data Presentation Modifications?**

Reviewer #1: (No Response)

Reviewer #2: (No Response)

**Summary and General Comments**

Reviewer #1: The study is interesting and the authors presented the information nicely. Some refinement as noted above would help with clarity.

Reviewer #2: Dear Authors, 

Thank you for the opportunity to review this interesting manuscript. The manuscript can potentially be published and might contribute to limited information in this field.

1. In the abstract the authors mentioned: STH infections have been associated with negative consequences for child physical and cognitive development and wellbeing. Thus, the aim of this study was to determine the association of STH infections in schoolchildren from Gqeberha, focusing on physical activity, physical fitness, and clustered cardiovascular disease (CVD) risk score.

-> Why it is important to check CVD risk score in the study?

The explanation of the connection has been mentioned in the background text. One short connecting sentence might need to be added in the abstract to make it a bit more clear. 

2. Regarding the background and the discussion, I am surprised that the authors did not come through studies by Pasha et al, Wiria et al and Tahapary et al, as they specifically concern on metabolic syndrome, cardiovascular risk (and atherosclerosis), and insulin sensitivity (resistance) in association with helminth infection and population at risk of epidemiological transition. They also did cohort study on this topic. They also have data on children and adult if that regarding some cardiovascular risk factors.

PLOS authors have the option to publish the peer review history of their article (what does this mean?). If published, this will include your full peer review and any attached files.

Reviewer #1: Yes: Helen Storey

Reviewer #2: Yes: Aprilianto Eddy Wiria, MD PhD
---

## [Decision Letter · Decision Letter 1]

15 Sep 2023

Dear Ms Nqweniso,

We are pleased to inform you that your manuscript 'Associations between soil-transmitted helminth infections and physical activity, physical fitness, and cardiovascular disease risk in primary schoolchildren from Gqeberha, South Africa.' has been provisionally accepted for publication in PLOS Neglected Tropical Diseases.

Best regards,

Sabine Specht

Academic Editor

Eva Clark

Section Editor

Reviewer's Responses to Questions

**Key Review Criteria Required for Acceptance?**

**Methods**

-Are the objectives of the study clearly articulated with a clear testable hypothesis stated?

-Is the study design appropriate to address the stated objectives?

-Is the population clearly described and appropriate for the hypothesis being tested?

-Is the sample size sufficient to ensure adequate power to address the hypothesis being tested?

-Were correct statistical analysis used to support conclusions?

-Are there concerns about ethical or regulatory requirements being met?

Reviewer #2: (No Response)

**Results**

-Does the analysis presented match the analysis plan?

-Are the results clearly and completely presented?

-Are the figures (Tables, Images) of sufficient quality for clarity?

Reviewer #2: (No Response)

**Conclusions**

-Are the conclusions supported by the data presented?

-Are the limitations of analysis clearly described?

-Do the authors discuss how these data can be helpful to advance our understanding of the topic under study?

-Is public health relevance addressed?

Reviewer #2: (No Response)

**Editorial and Data Presentation Modifications?**

Reviewer #2: (No Response)

**Summary and General Comments**

Reviewer #2: Thank you very much for the revision of the manuscript. It has been improved than the previous version.

I hope this can be addition to the published knowledge in the field of helminth infections and cardiovascular diseases.

PLOS authors have the option to publish the peer review history of their article (what does this mean?). If published, this will include your full peer review and any attached files.

Reviewer #2: **Yes: **Aprilianto Eddy Wiria, MD PhD

---

## [Editor Report · Acceptance letter]

28 Sep 2023

Dear Ms Nqweniso,

We are delighted to inform you that your manuscript, "Associations between soil-transmitted helminth infections and physical activity, physical fitness, and cardiovascular disease risk in primary schoolchildren from Gqeberha, South Africa.," has been formally accepted for publication in PLOS Neglected Tropical Diseases.

Best regards,

Shaden Kamhawi

co-Editor-in-Chief

Paul Brindley

co-Editor-in-Chief
